# Finite Element Analysis of Renewable Porous Bones and Optimization of Additive Manufacturing Processes

**Hailong Ma [1], Shubo Xu [1,*], Xiaoyu Ju [1], Aijun Tang [2] and Xinzhi Hu [1]**

[1] School of Materials Science and Engineering, Shandong Jianzhu University, Jinan 250101, China; mahailong@sdjzu.edu.cn (H.M.); wuqiangdavid@163.com (X.J.); 12285@sdjzu.edu.cn (X.H.)

[2] School of Mechanical and Electrical Engineering, Shandong Jianzhu University, Jinan 250101, China; tajsmile@sdjzu.edu.cn

\* Correspondence: xsb@sdjzu.edu.cn

**Abstract:** Three-dimensional printing technology has a precise manufacturing process that can control tiny pores and can design individualized prostheses based on the patient's own conditions. Different porous structures were designed by controlling different parameters such as porosity, using UG NX to establish models with different porosities and using ANSYS to simulate stress and strain. Unidirectional compression and stretching simulations were carried out to obtain stress, strain, and deformation. Based on these data, a porosity was found to approximate the elastic modulus of the humeral bone scaffold. As the porosity increased, the equivalent elastic modulus decreased significantly in the lateral direction, and the maximum stress formed by the porous structure and deformation increased significantly. Four different finite element models and geometric models of cubic, face-centered cubic, honeycomb, and body-centered cubic unit structures were selected. Then these porous structures were simulated for tensile and compression experiments, and the simulation results were analyzed. The forming simulation of the finite element model was carried out, and the evolution of mechanical properties of the porous structure during the 3D printing process was analyzed. The results showed that designing the humeral bone scaffold as a porous structure could reduce the stiffness of the prosthesis, alleviate stress shielding around the prosthesis after surgery, enhance its stability, and prolong its service life. The study provides reference values and scientific guidance for the feasibility of porous humeral bone scaffolds and provides a basis for the research and design of clinical humeral bone scaffolds.

**Keywords:** bone scaffold; finite element analysis; compression experiment; additive manufacturing; porosity





## 1. Introduction

Three-dimensional printing is a technology that uses additive manufacturing devices to layer specialized materials based on a pre-planned three-dimensional model, to produce the desired parts [1–3]. Compared with traditional manufacturing techniques, 3D printing does not require complex forging processes to obtain the final desired parts [4]. The mechanical properties of porous metal materials and the bone tissue they replace are well matched, with rough inner and outer surfaces providing good adhesion for bone cells and allowing for the necessary nutrients and moisture to move through interconnected pores [5–7]. The conventional design methods have obvious technical limitations and cannot achieve the fine parameter requirements for pore size and complex pore morphology [8–10]. Additionally, designing complex irregular porous structural components is also challenging [11]. However, the use of 3D printing technology in computer-aided design can be utilized to create models of porous metals with complex pore morphologies and fine parameters, thus enabling the realization of the characteristics of human bone scaffolds [12]. Porous structured materials have a wide range of applications and include good mechanical

and physical properties. However, controlling the porous structure during production is difficult, which can result in porous structures that have some shortcomings in terms of their properties [13]. The application and development of 3D printing technology provides new directions in the preparation of porous structures [14,15].

By using selective laser melting (SLM), laser engineered net shaping (LENS), and selective laser sintering (SLS), the micro and macro pores of the material can be understood by controlling the powder sintering degree and laser scanning trajectory [16,17]. SLM is generally used in the preparation of metal porous structures, and SLS can be used to prepare many types of porous materials. The porous structure prepared by SLM is more precise and detailed than that produced by other methods and can concentrate the energy of the electron beam and laser [18]. Artificial bone scaffolds act as templates for the formation of extracellular matrix and cell interactions in bone, providing structural support for newly formed tissue and are a type of transplantable bone material used for repairing bone defects [19]. In the design stage of artificial bone scaffolds, suitable performance expression parameters, design methods, and manufacturing methods are used to design scaffold structures to meet their mechanical and biological performance requirements, including mechanical compression performance as well as the biological properties of pore connectivity and biological permeability [3].

Analysis of the specific functional requirements and design standards of the artificial humeral bone scaffold is necessary to determine the appropriate structural design for mechanical performance (mechanical compression performance) and biological performance (porosity connectivity and biological permeability), providing a foundation for the design of the artificial bone scaffold structure [20–24]. The pore size and porosity range for the 316 L porous scaffold are determined based on the practical environment, conditions, and requirements for the human body as well as relevant literature and studies applied to different parts of the human body [25–28]. Different porous scaffold tissue and performance analyses are conducted for different porosity rates. Using UG NX to create models of different porosity rates, ANSYS is used to simulate stress-strain, and different porous models are designed by controlling different parameters such as porosity rate. Then unidirectional compression simulation is conducted to obtain stress, strain, and deformation data that are analyzed to obtain the porosity rate closely matching the elastic modulus of the humeral bone scaffold [29,30].

## 2. Porous Structure Model

### 2.1. Design of Porous Structure

The structural shape of the unit cell is an important factor influencing the mechanical performance of the porous structure and affects the internal layout of the porous structure. Based on a large amount of research on unit cell structures and considering the current level of additive manufacturing, this study designed four simple and non-twisted porous structures using the UG NX modelling software, version 11.0, as shown in Figure 1: (a) cubic unit cell structure, (b) face-centered cubic unit cell structure, (c) body-centered cubic unit cell structure, and (d) honeycomb unit cell structure.

Pore size, porosity, strut diameter, strut length, and external dimensions of the porous structure are the main parameters for the design of the porous structure. These parameters play a crucial role in the application and mechanical performance of the designed bone scaffold. When designing a porous structure, the following aspects should be considered: first, the designed porous structure should have connectivity to meet the supply of nutrients and discharge of metabolic waste in the human body; second, the designed porous structure must have excellent mechanical properties, especially with a certain strength and stiffness, to withstand external impact and tension. Even when subjected to external forces, it can still maintain high mechanical performance.

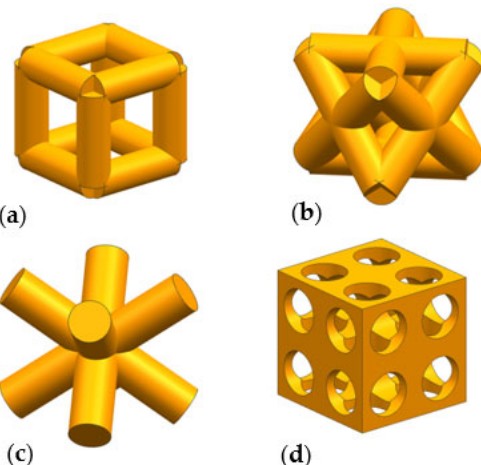

**Figure 1.** Different types of porous scaffold structures: (**a**) cubic unit cell structure, (**b**) face-centered cubic unit cell structure, (**c**) body-centered cubic unit cell structure, and (**d**) honeycomb unit cell structure.

### 2.1.1. Design of Strut Diameter and Length

The diameter and length of the stubs are shown in Figure 2. The sizes of strut diameter *d* and length *l* directly affect the size of the porous structure, the overall volume of the porous structure, and the solid volume, thereby indirectly affecting the size of the porosity. In other words, to control the porosity of the porous structure, the diameter and length of the struts can be adjusted. Based on comprehensive considerations of the size and manufacturing angle of the human humeral bone cross-section, the strut diameter is usually controlled between 200 and 700 μm, and the length is usually 1–2 mm. In this study, the diameter and length of the struts were varied in order to achieve the desired porosity rate, with the strut diameter controlled between 250 and 700 μm and the strut length set to 1 mm.

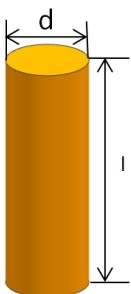

**Figure 2.** Strut diameter and length. d is the diameter of the stanchion and l is the length.

### 2.1.2. Porosity

The porosity of a porous medium refers to the ratio of the total volume of small voids within the porous structure to the apparent total volume of the porous structure. It is related to the diameter and length of the struts and the unit cell structure. The calculation formula for the porosity of a porous structure is as follows:

$$\Phi = \frac{V_h}{V_t} \times 100\% = \frac{V_t - V_s}{V_t} \times 100\% \ (2.1) \tag{1}$$

where $\Phi$ is the porosity of the porous structure, $V_h$ is the pore volume (mm$^3$), $V_t$ is the total volume of the structure (mm$^3$), and $V_s$ is the volume of the solid part (mm$^3$).

The US Food and Drug Administration (FDA) recommends a porosity range of 30%–70% [13] for porous femoral implants. Bragdon et al., suggested that the porosity of a porous structure optimized for bone growth should be at least 40% [14]. Simoneau et al., suggested that the porosity of a porous structure should be between 30% and 50% [15–19].

In this study, four different types of porous structures were designed, and by adjusting the strut diameter and length, two porosity levels were set for each structure. The high porosity level was around 86%, and the low porosity level was around 65%.

### 2.1.3. Design Parameters

This study designed two variables to analyze the differences in the mechanical properties of the porous structures. The two variables were porosity and porous structure. By controlling two porosity levels and designing four porous structures, eight models were formed for comparison.

First, four types of porous structures were defined with a high porosity range of 80%–90% and a low porosity range of 60%–70%. Based on the porosity range, the strut diameter was set between 250 and 700 μm [20]. The edge length of the porous structures was measured using the "distance" command in UG NX 11.0 software, and the solid volume was measured using the "measure volume" command. The porosity was then calculated using formula 2.1, and it was verified whether the porosity was within the required range. The design parameters for the cubic, face-centered cubic, body-centered cubic, and honeycomb porous structures are shown in Table 1.

**Table 1.** Design parameters for porous structures.

|  | Porous Structure | Strut Diameter (μm) | Pore Size (μm) | Porosity (%) |
|---|---|---|---|---|
| High Porosity Level | Cubic | 350 | 650 | 85 |
|  | Face-centered cubic | 350 | 650 | 85 |
|  | Honeycomb | 450 | 450 | 85 |
|  | Body-centered cubic | 200 | 620 | 85 |
| Low Porosity Level | Cubic | 700 | 300 | 65 |
|  | Face-centered cubic | 400 | 600 | 65 |
|  | Honeycomb | 350 | 350 | 65 |
|  | Body-centered cubic | 350 | 470 | 65 |

### 2.1.4. Material

One of the commonly used medical metal materials is 316 L stainless steel, which has benefits such as good machinability, high strength, low cost, and strong corrosion resistance [26–28]. It is currently used in clinical medicine as a bio-stent made of 316 L stainless steel, which has mechanical properties similar to human bone tissue and good biocompatibility. Traditional manufacturing techniques cannot meet the market demand for complex shapes and various sizes of artificial joints, while the SLM technology can produce complex stents and has been increasingly used in the field of biomedicine. The material selected in this study is 316 L stainless steel, which has high strength and good mechanical properties.

The 316 L stainless steel powder has a spherical shape, and the powder diameter is about 40 μm. The composition of the 316 L stainless steel material is shown in Table 2, and the mechanical properties are shown in Table 3. The 3D printer used to manufacture porous structures in SS 316 L is the FS121M metal additive manufacturing machine developed by Hunan Huashu High-Tech Company Limited, Changsha City, China. The parameters of 3D printing equipment are: laser power 150 W, scanning speed 500 mm/s, scanning distance 0.06 mm, and powder layer thickness 0.035 mm.

**Table 2.** Composition of 316 L stainless steel.

| Grade | C | Si | Mn | P | S | Ni | Cr | Mo |
|---|---|---|---|---|---|---|---|---|
| 00Cr17NiMo2 | ≤0.03 | ≤1.00 | ≤2.00 | ≤0.035 | ≤0.03 | ≤12.0–15.0 | ≤16.0–18.0 | ≤2.0–3.0 |

**Table 3.** Mechanical properties of 316 L stainless steel.

| Property | Yield Strength (MPa) | Tensile Strength (MPa) | Elongation (%) | Vickers Hardness (Hv) | Remarks |
|---|---|---|---|---|---|
| Standard | ≥205 | ≥520 | ≥40 | ≤200 | 2 B/1.5 t |
| Typical | 310 | 620 | 53 | 155 | |

### 2.2. Porous Structure Models

According to the design parameters mentioned above, the designed unit cells were used to construct four types of porous structures using UG NX 11.0 modeling software as shown in Figure 3.

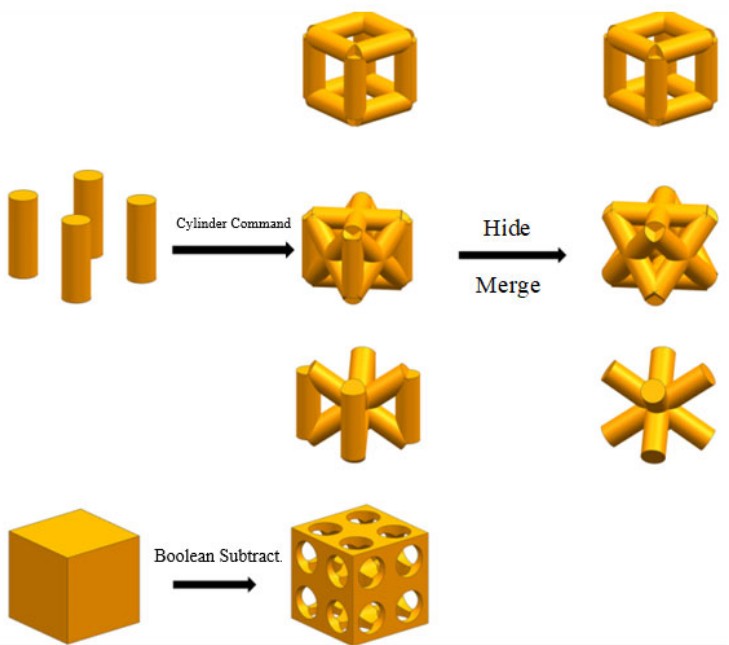

**Figure 3.** Modeling process for unit cell structures.

Four different types of unit cells were created using the designed parameters. The porous structures were established using the array geometry feature options and Boolean Operator commands, and the resultant porous structures' geometric models are shown in Figure 4. Geometry operations for the four porous structure models were checked. The volume check included boundary, consistency, data structure, and surface intersection; the surface check included spikes/cuts, smoothness, and self-intersection; the edge check included smoothness and tolerance. After the four porous structures were checked, all four were shown to pass the geometry check.

Four different types of unit cell structures were designed in terms of support column diameter, pore size, and pore volume ratio. First, the support column diameter that meets the requirements of 3D printing technology was designed, as were pore sizes that facilitate the transport of nutrients and metabolic waste in cells and are favorable for bone tissue ingrowth. Four different types of unit cell structures, namely cubic, face-centered cubic, honeycomb, and body-centered cubic structures, were established. The four types of unit cell structures were arranged in an orderly manner in 3D space to form the geometric models of the cubic, face-centered cubic, honeycomb, and body-centered cubic porous structures, respectively. Finite element models were established based on the geometric models of the four types of porous structures.

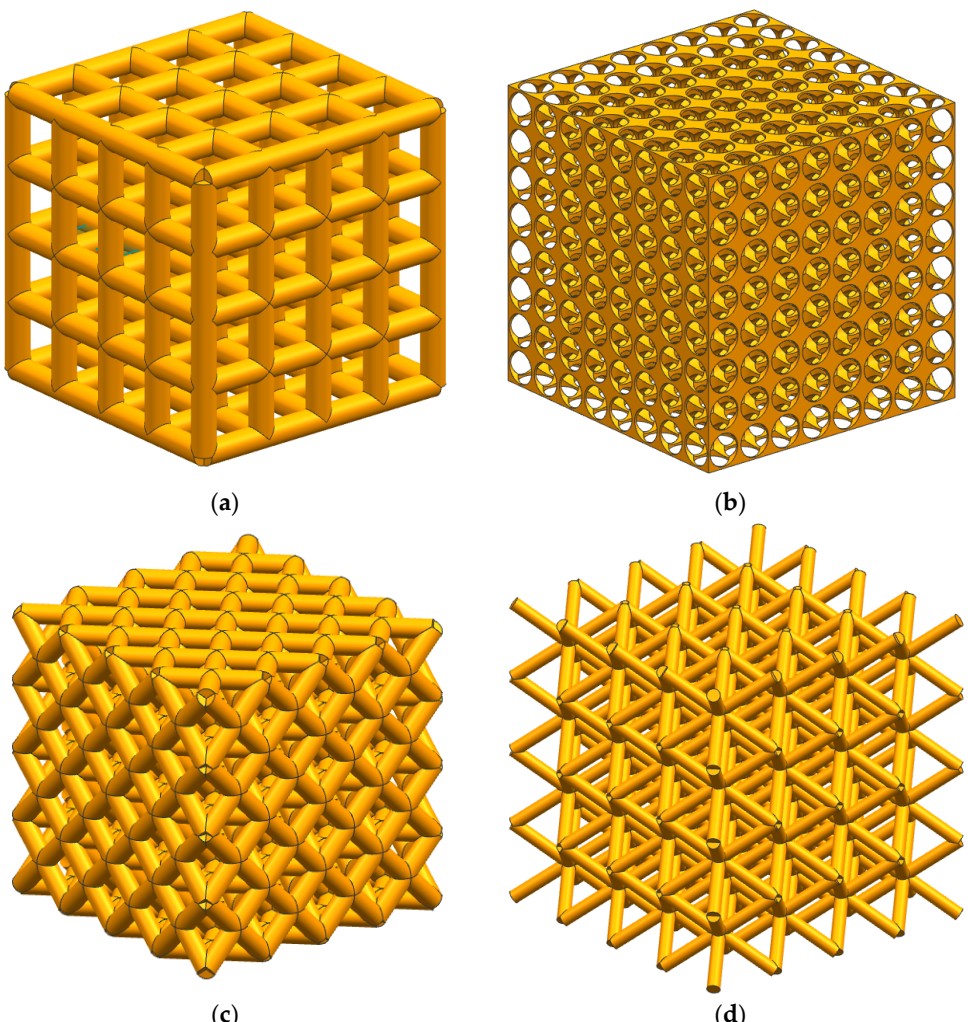

**Figure 4.** Geometric models of different porous structures: (**a**) cubic porous structure, (**b**) honeycomb porous structure, (**c**) face-centered cubic porous structure, and (**d**) body-centered cubic porous structure.

## 3. Finite Element Simulation of the Porous Scaffold

Finite element simulation of the porous scaffold was carried out using the ANSYS Workbench software (version17.0) module for unidirectional compression and tensile calculation. Static mechanical performance analysis was conducted using adaptive grid division by ANSYS. In the material library, 316 L stainless steel was selected. In the mesh-method-geometry structure option, the porous structure to be meshed was selected, and the method was changed to tetrahedron automatically. In the mesh-size adjustment-geometry structure option, the mesh size was set to 0.005 mm to increase the calculation accuracy and save computational time.

Different load forces and load directions directly or indirectly act on the model structure, resulting in stress and deformation inside the structure. Different load and load direction choices were made in the study because of different applied loads and load orientations. The humerus bone can be assumed to be lifting a weight when subjected to tension. A tensile force of 1000 N was specified, assuming the human body can withstand a force equivalent to carrying a 90.72 Kg object. The humerus bone can be assumed to be doing push-ups when subjected to compression. A compressive force of 1000 N was specified, assuming the human body weighs 90.72 Kg. Therefore, compression and tension experiments only required changing the direction of the applied force.

### 3.1. Compression Analysis

Figure 5a shows the simulation of the compression stress of the cubic porous structure with a porosity of 65%. It can be seen from Figure 5a that the stress concentration phenomenon of the cubic structure mainly occurs near the columns and connection points, while the transverse columns produce smaller forces. The reason for this stress concentration is that the direction of the compression force is vertical, and the cubic structure is composed of vertical and horizontal columns. The vertical columns bear more force in the vertical direction, while the horizontal columns bear less. Figure 5b shows the simulation of the compression stress of the face-centered cubic porous structure with a porosity of 65%. It can be seen from the Figure 5 that the stress concentration phenomenon of the face-centered cubic structure mainly occurs on the four sides of the unit cell, and the stress on the upper and lower surfaces of the unit cell is smaller. The reason for this stress concentration is that the porous scaffold is obtained by an array unit structure, and there will be certain defects in the contact points of the unit columns, resulting in stress concentration.

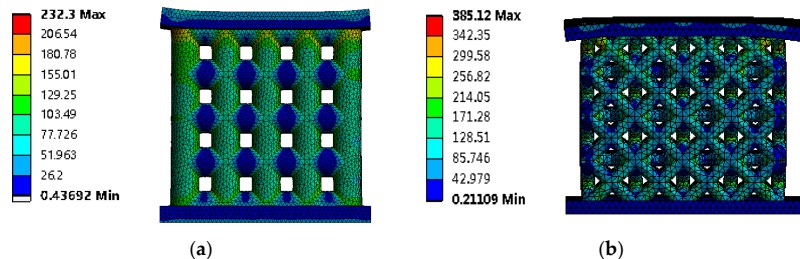

(**a**) (**b**)

**Figure 5.** (**a**) The compression stress cloud map of the cubic porous structure with a porosity of 65%. (**b**) The compression stress cloud map of the face-centered cubic porous structure with a porosity of 65%.

Figure 6a shows the simulation of the compression stress of the honeycomb porous structure with a porosity of 65%. It can be seen from Figure 6a that the stress concentration phenomenon of the face-centered cubic structure mainly occurs near the holes, especially at the outer edge of the porous structure. The stress between adjacent holes in the vertical direction is smaller. The holes close to the applied force appear significantly deformed, showing irregular circular shapes. The reason for this is that the outer wall of this porous structure is relatively thin, and spherical powder particles can be seen around the holes, indicating incomplete sintering. Some concave edges are caused by excessive melting. Figure 6b shows the simulation of the compression stress of the body-centered cubic porous structure with a porosity of 65%. It can be seen from Figure 6b that the stress concentration phenomenon of the body-centered cubic structure mainly occurs near the connecting holes of the scaffold, and the stress on the cylinders is smaller. The reason for this stress concentration is that the porous scaffold is obtained by an array unit structure, and there will be certain defects in the contact points of the unit columns, resulting in stress concentration.

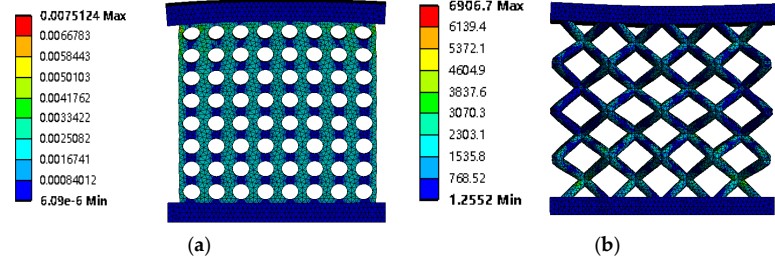

(**a**) (**b**)

**Figure 6.** (**a**) The compression stress cloud map of the honeycomb porous structure with a porosity of 65%. (**b**) The compression stress cloud map of the body-centered cubic porous structure with a porosity of 65%.

*3.2. Compression Analysis of Low-Porosity Structures*

By applying a uni-axial compression to the top of the porous scaffold, strain cloud maps and deformation maps of the different porous scaffold structures were obtained. Figure 7 shows the strain maps of four low-porosity structures, and Figure 8 shows the deformation maps of four low-porosity structures under compression experiments. Comparing the deformation maps of the four structures under compression experiments, it can be seen that the shape and unit structure of the porous structure underwent significant deformation compared to their original shape, and the pore size underwent severe deformation. In particular, during the compression process of the face-centered cubic structure, its deformation trend was to expand laterally from the middle part, and the final shape tended to be a mostly bulging round shape in the lateral central axis direction. As the simulation method used here is numerical analysis, fracture and failure phenomena cannot be easily observed. However, the values and distributions of the data can be used to indicate such phenomena. Table 4 shows the average strain and average deformation of the four structures. It can be observed from Table 4 that the body diagonally arranged structure has better mechanical properties in terms of strain, while the cubic structure has the smallest deformation under compression.

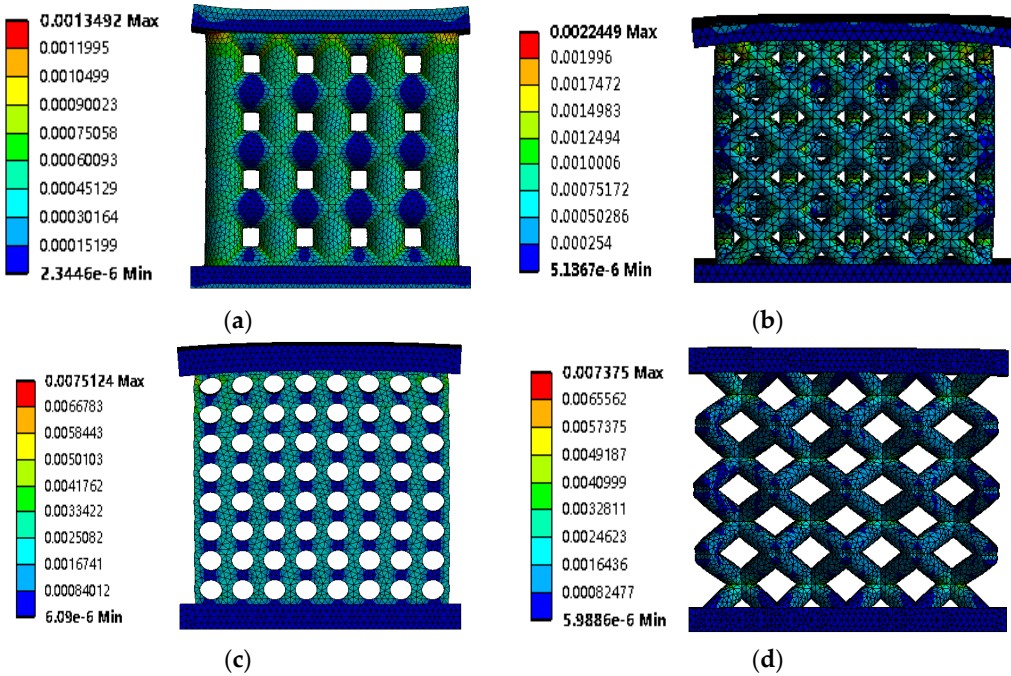

**Figure 7.** The strain maps of four low-porosity structures under compression experiments. (**a**) The strain map of the cubic porous structure with a porosity of 65%. (**b**) The strain map of the face-centered cubic porous structure with a porosity of 65%. (**c**) The strain map of the honeycomb porous structure with a porosity of 65%. (**d**) The strain map of the body-centered cubic porous structure with a porosity of 65%.

**Table 4.** The strain and deformation of the four low-porosity porous structures under compression experiments.

|  | Cubic | Body-Centered Cubic | Honeycomb | Face-Centered Cubic |
|---|---|---|---|---|
| Strain (mm) | 0.000312 | 0.001013 | 0.000841 | 0.000438 |
| Deformation (mm) | 0.000968 | 0.008143 | 0.003922 | 0.002768 |

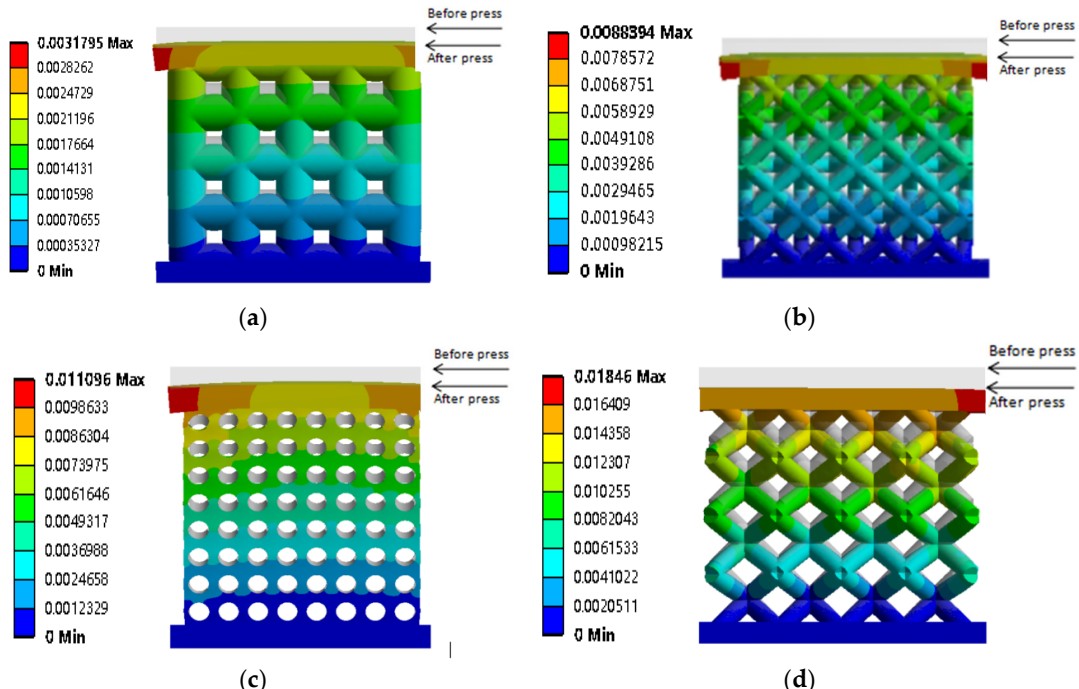

**Figure 8.** The deformation maps of these four low-porosity structures under compression experiments. (**a**) The deformation map of the cubic porous structure with a porosity of 65%. (**b**) The deformation map of the face-centered cubic porous structure with a porosity of 65%. (**c**) The deformation map of the honeycomb porous structure with a porosity of 65%. (**d**) The deformation map of the body-centered cubic porous structure with a porosity of 65%.

### 3.3. Compression Analysis of High-Porosity Structures

Figure 9a shows the compression stress cloud map of the cubic porous structure with a porosity of 85%. It can be seen from Figure 9a that the stress concentration phenomenon of the cubic structure mainly occurs near the columns and connecting points, while the transverse beams produce smaller forces. Its stress concentration phenomenon is basically the same as that of the same structure with a low porosity, only the range is slightly larger. Figure 9b shows the compression stress cloud map of the face-centered cubic porous structure with a porosity of 85%. Its stress concentration phenomenon is basically the same as that of the same structure with a low porosity, only the range is slightly larger.

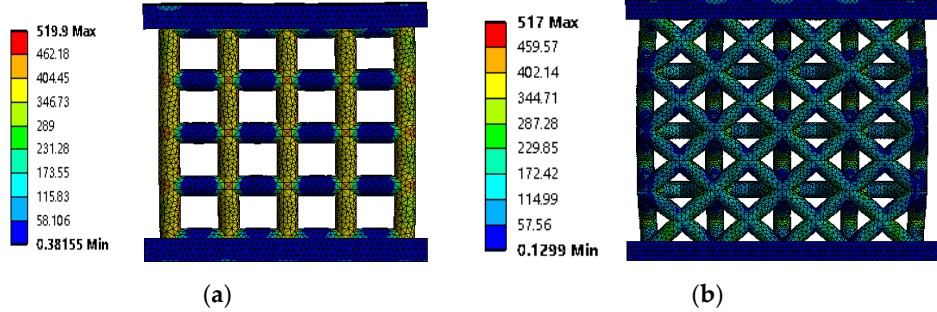

**Figure 9.** (**a**) The compression stress cloud map of the cubic porous structure with a porosity of 85%. (**b**) The compression stress cloud map of the face-centered cubic porous structure with a porosity of 85%.

Figure 10a shows the compression stress cloud map of the honeycomb porous structure with a porosity of 85%. It can be seen from Figure 10a that the stress concentration of the honeycomb structure mainly occurs near the pores, especially around the periphery of the

porous structure. The stress between adjacent pores in the longitudinal direction is small, and the pores near the applied force deform significantly, appearing as irregular circles. Its stress concentration phenomenon is basically the same as that of the same structure with a low porosity, but the range is slightly smaller. Figure 10b shows the compression stress cloud map of the body-centered cubic porous structure with a porosity of 85%. As can be seen from Figure 10b, the stress concentration of the body-centered cubic structure mainly occurs near the connecting parts around the holes of the bracket. The stress on the cylinder is smaller, and its stress concentration phenomenon is basically the same as that of the same structure with a low porosity, only the range is slightly larger.

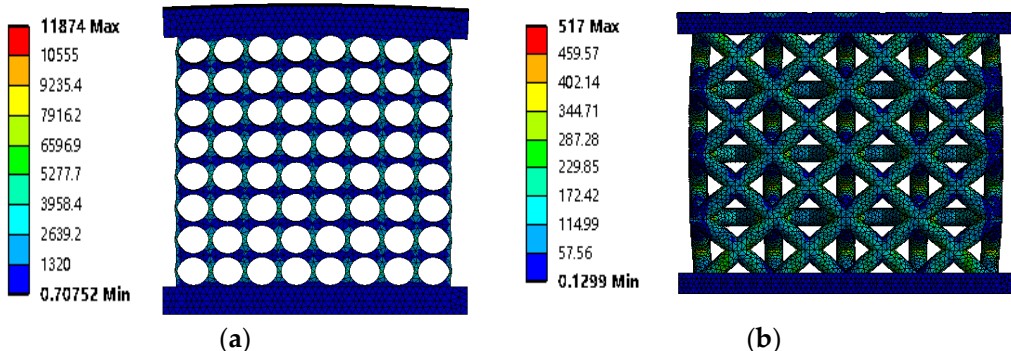

(**a**)             (**b**)

**Figure 10.** (**a**) The compression stress cloud map of the honeycomb porous structure with a porosity of 85%. (**b**) The compression stress cloud map of the body-centered cubic porous structure with a porosity of 85%.

A uni-axial force was applied to the top of the porous scaffold to obtain the strain and deformation maps of different porous scaffold structures. Figure 11 shows the strain maps of four high-porosity structures, and Figure 12 shows the deformation maps of the compression experiments of the four structures. By comparing the deformation maps of the four structures in the compression experiments, it can be seen that the shape and unit structure of the porous structures have undergone significant deformation compared to the original shape, and the pore size has undergone severe morphological deformation, especially the face-centered cubic structure. During the compression process, its deformation trend is the horizontal expansion of the middle part of the structure, and the final shape tends to be a horizontally central axis with a maximum bulge. Since the simulation uses numerical analysis methods, the fracture mechanism cannot be intuitively observed, but it can be identified by the size and distribution of data values. The average strain and deformation values of the four structures are shown in Table 4. From Table 5, it can be seen that the cubic porous structure has better strain mechanical properties and the smallest deformation after compression.

**Table 5.** The strain and deformation values of the four high-porosity porous structures in the compression experiments.

|  | Cubic | Body-Centered Cubic | Honeycomb | Face-Centered Cubic |
|---|---|---|---|---|
| Strain (mm) | 0.000704 | 0.003176 | 0.002791 | 0.005581 |
| Deformation (mm) | 0.004098 | 0.072260 | 0.023360 | 0.041275 |

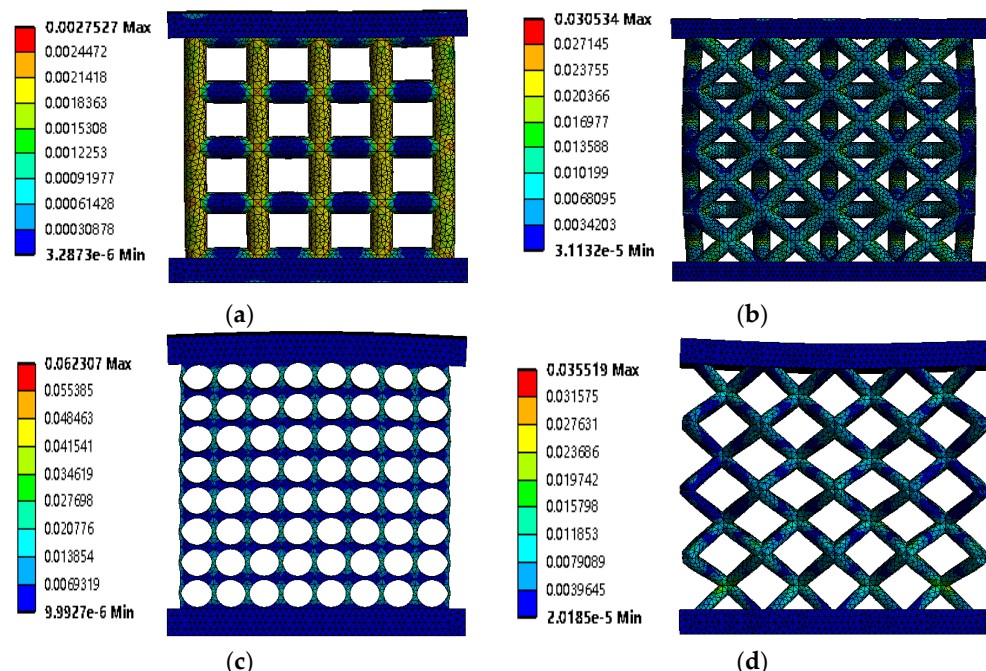

**Figure 11.** The strain maps of the four high-porosity structures in the compression experiments. (**a**) Strain map of the cubic porous structure with a porosity of 85%. (**b**) Strain map of the face-centered cubic porous structure with a porosity of 85%. (**c**) Strain map of the honeycomb porous structure with a porosity of 85%. (**d**) Strain map of the body-centered cubic porous structure with a porosity of 85%.

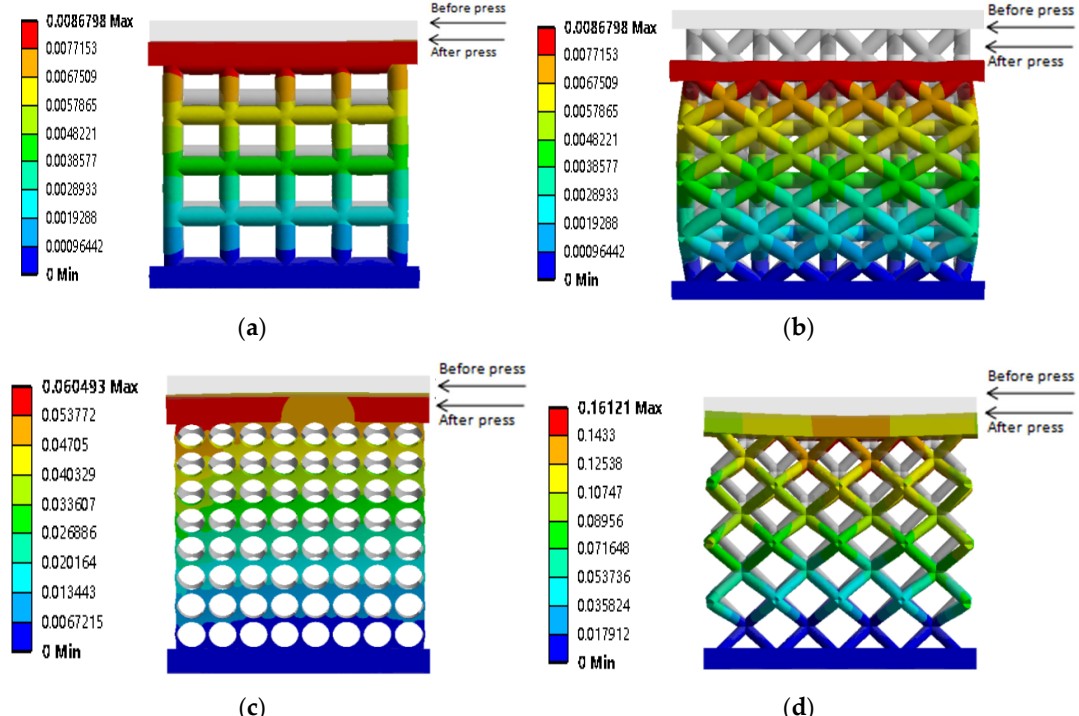

**Figure 12.** The deformation maps of the compression experiments of the four high-porosity structures. (**a**) Deformation map of the cubic porous structure with a porosity of 85%. (**b**) Deformation map of the face-centered cubic porous structure with a porosity of 85%. (**c**) Deformation map of the honeycomb porous structure with a porosity of 85%. (**d**) Deformation map of the body-centered cubic porous structure with a porosity of 85%.

Table 6 shows the stress and strain values of high and low porosity obtained from compression simulations of different porous structures. By comparing and analyzing the data in Table 6, it can be seen that the maximum stress formed in the structure of the porous material increases significantly with an increase in porosity. This is consistent with the results of the calculation because as the porosity increases while the pore size remains constant, the distance between the pores decreases significantly, resulting in thinning of the microstructures of the pore walls and a change in the structural strength. Overall, for the porous model with material properties assigned as 316 L stainless steel, porosity is the main factor influencing its mechanical properties.

**Table 6.** The stress and strain values of high and low porosity obtained from compression simulations of different porous structures.

| Porous Structures / Porosity | | Cubic | Body-Centered Cubic | Honeycomb | Face-Centered Cubic |
|---|---|---|---|---|---|
| Stress values (MPa) | high | 132.88 | 103.61 | 488.42 | 561.52 |
| | low | 59.52 | 75.422 | 152.58 | 178.75 |
| Strain values (mm) | high | 0.000704 | 0.002791 | 0.002791 | 0.003318 |
| | low | 0.000318 | 0.000841 | 0.000841 | 0.001013 |

By comparing and analyzing the stress, strain, and deformation of the four structures, it can be concluded that, in terms of stress concentration during compression, the cubic porous structure is the best. In terms of a combination of strain and deformation, the cubic porous structure exhibited the best mechanical properties.

## 4. Porous Bracket Forming Simulation

### 4.1. Stress Analysis of Low-Porosity Porous Structures in Additive Manufacturing

Figure 13a shows the stress distribution of the cubic porous structure with low porosity after the forming simulation. It can be seen from the Figure 13a that the residual stress concentration mainly occurs at the crossbar and inner column, and the stress on the outer surface and bottom of the porous structure is relatively small. Figure 13b shows the stress distribution of the honeycomb porous structure with low porosity after the forming simulation. It can be seen that the residual stress mainly concentrates on the upper part of the hole and the outer periphery of the porous structure, and small forces are generated on the bottom and inside of the porous structure. The reason for this phenomenon may be that uneven heating occurs due to the heat input during welding, causing some areas to be overheated, resulting in the melting of the welding gap. The material next to the melting pool, which is subjected to higher temperatures, is constrained by nearby materials and forms an uneven compressive plastic deformation during cooling. When the materials that have undergone plastic deformation are constrained by surrounding materials, they cannot contract actively, resulting in tensile stress; after the weld metal is completely cooled, obstacles to smooth contraction can also result in corresponding tensile stress.

Figure 14a illustrates the stress distribution during the formation process of a low-porosity face-centered cubic porous structure. Residual stresses mainly concentrate at the center of the porous structure and at the interface between the porous structure and the substrate. The initial temperature of the substrate also affects the residual stress distribution. Figure 14b clearly depicts the existence of large residual stresses in the support before the substrate is removed. Such stresses concentrate on the bottom and are caused by the construction process, which initially involves a large area of construction and a relatively small number of powder layers. The high-energy laser beam burns the substrate, resulting in increased thermal stress. In addition, SLM's rapid cooling and heating characteristics cause constrained shrinkage of the support structure at the bottom, leading to a large strain and residual stress. However, this phenomenon is effectively alleviated after the substrate

is removed, resulting in a reduced residual stress in the completed structure as shown in Figure 14c.

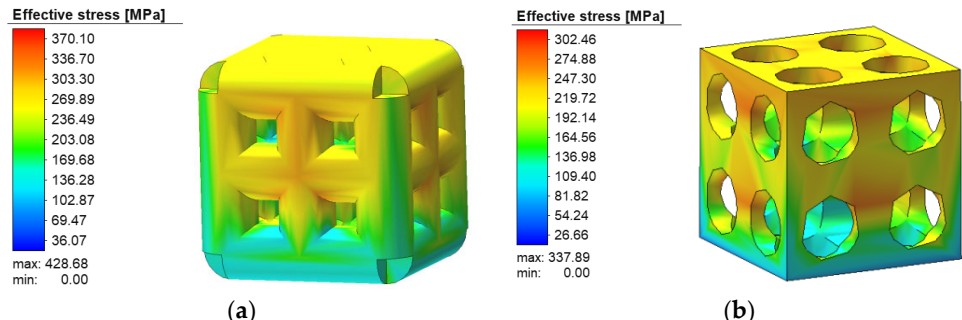

**Figure 13.** (**a**) Stress cloud image of the cubic porous structure with a porosity of 65%. (**b**) Stress cloud image of the honeycomb structure with a porosity of 65%.

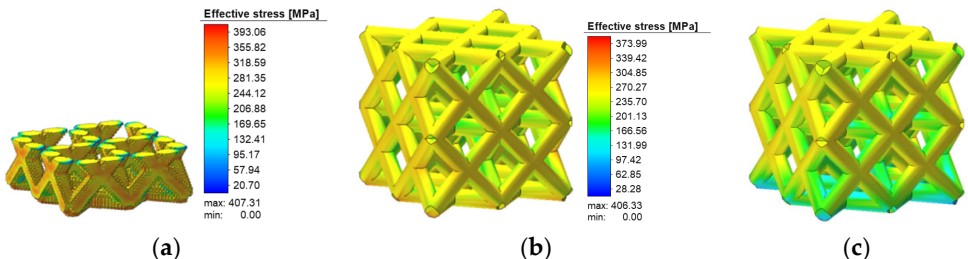

**Figure 14.** Stress cloud images of a 65% porosity face-centered cubic porous structure (**a**) during the formation process, (**b**) before the substrate is removed, and (**c**) after the substrate is removed.

Figure 15a shows the stress distribution of a low-porosity body-centered cubic porous structure, with residual stresses mainly concentrated at the center of the porous structure and at the interface between the porous structure and the substrate. After the completed structure is cut off from the substrate as shown in Figure 15c, the residual stress at the interface between the porous structure and the substrate is released due to the relaxation of stress. The residual stresses in both the face-centered cubic and body-centered cubic porous structures are caused by the increased thermal stress and the constraints from the lower layers during the construction process with a large area, particularly at the beginning of the construction.

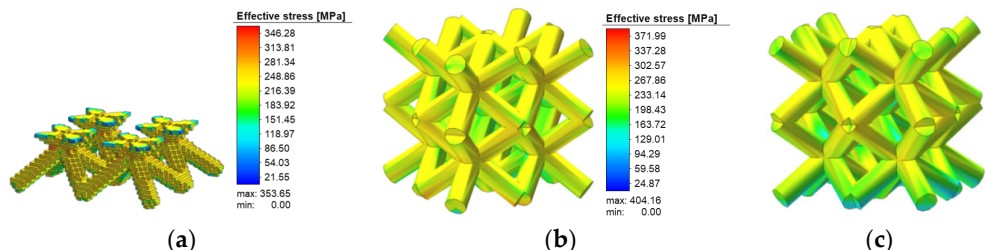

**Figure 15.** Stress cloud images of a 65% porosity body-centered cubic porous structure (**a**) during the formation process, (**b**) before the substrate is removed, and (**c**) after the substrate is removed.

Table 7 shows the simulated stress values of four low-porosity porous structures. It can be seen from Table 7 that the honeycomb porous structure has the smallest residual stress after construction, resulting in improved fatigue performance and better mechanical properties. This allows the bone scaffold to withstand certain impact forces, and it has higher strength, hardness, and processability.

**Table 7.** Simulated stress values of four low-porosity porous structures.

| Porous Structures | Cubic | Face-Centered Cubic | Honeycomb | Body-Centered Cubic |
|---|---|---|---|---|
| Stress values (MPa) | 370.10 | 373.99 | 302.46 | 371.99 |

*4.2. Analysis of Stress Results for Additively Manufactured High-Porosity Porous Structures*

Figure 16a shows the stress simulation image of a high-porosity cubic porous structure. It can be seen from Figure 16a that residual stresses after construction mainly concentrate in the crossbars and inner pillars of the cubic structure, while the stress on the outer surface and bottom of the porous structure is relatively small. The concentration of residual stress after construction is basically the same as the low-porosity structure, with only a slightly larger range. Figure 16b shows the stress simulation image of a high-porosity honeycomb porous structure. It can be seen from Figure 16b that residual stresses after construction mainly concentrate in the upper part of the holes and the periphery of the porous structure, while the stress on the bottom and inside of the porous structure is relatively small. The concentration of residual stress after construction is basically the same as the low-porosity structure, with only a slightly larger range.

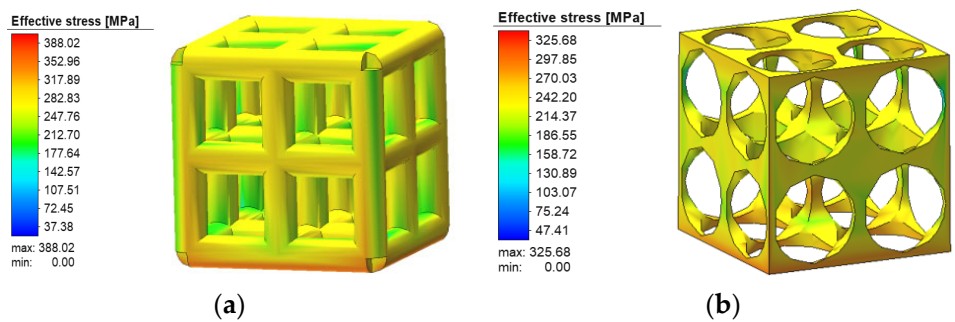

**Figure 16.** (**a**) Stress cloud images of an 85% porosity cubic porous structure during the formation process. (**b**) Stress cloud images of an 85% porosity honeycomb cubic porous structure during the formation process.

Figure 17a shows the stress simulation image of a high-porosity face-centered cubic porous structure during the formation process. It can be seen from Figure 17a that residual stresses after construction mainly concentrate at the center of the unit cells and where the porous structure bottom contacts the substrate. The concentration of residual stress after construction is basically the same as the low-porosity structure, with only a slightly larger range. Figure 17b shows the stress simulation image of a high-porosity body-centered cubic porous structure. It can be seen from Figure 17b that residual stresses after construction mainly concentrate at the center of the unit cells and where the porous structure bottom contacts the substrate. The concentration of residual stress after construction is basically the same as the low-porosity structure, with only a slightly larger range.

The phenomenon of mass movement due to a gradient of tension between the interfaces of two liquids with different surface tensions is known as the Marangoni effect. The energy density that can be absorbed by the metal material during construction easily induces a strong Marangoni effect, which can easily form cracks at the weld seam while adding to the risk of stress corrosion after implantation, as the human blood is weakly alkaline, which can accelerate stress corrosion near the weld seam.

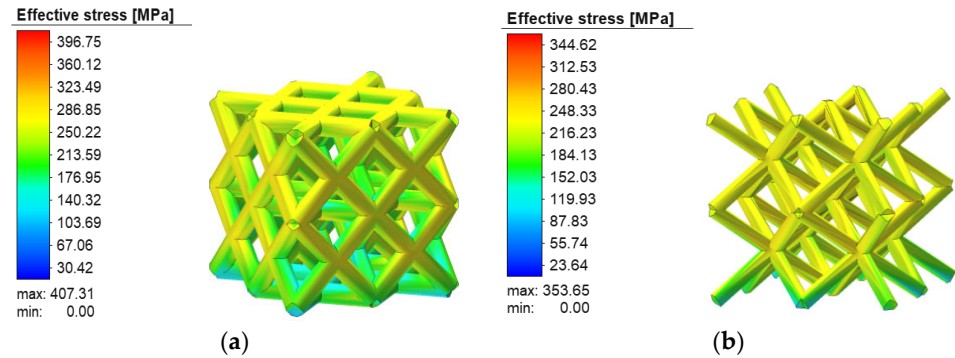

**Figure 17. (a)** Stress cloud images of an 85% porosity face-centered cubic porous structure during the formation process. **(b)** Stress cloud images of an 85% porosity body-centered cubic porous structure during the formation process.

Table 8 shows the stress values of the four types of low-porosity porous structures during the formation simulation. It can be seen from Table 8 that the honeycomb porous structure has the smallest residual stress after construction, which leads to an improvement in its fatigue performance, and thus has better mechanical properties, enabling the scaffold to withstand some impact and possessing high strength, hardness, and machinability.

**Table 8.** Stress values of the four types of low-porosity porous structures during the formation simulation.

| Porous Structures | Cubic | Face-Centered Cubic | Honeycomb | Body-Centered Cubic |
|---|---|---|---|---|
| Stress values (MPa) | 388.02 | 396.75 | 325.68 | 344.62 |

Calculation of the residual stress values after construction for the high- and low-porosity porous structures can be obtained from the formation simulation results. For the cubic porous structure, the residual stress after construction for the high- and low-porosity structures are 388.02 Pa and 370.10 MPa, respectively, showing a decrease of 4.6% in the high-porosity structure. For the face-centered cubic porous structure, the residual stress after construction for the high- and low-porosity structures are 396.75 MPa and 373.99 Pa, respectively, showing a decrease of 5.7% in the high-porosity structure. For the honeycomb porous structure, the residual stress after construction for the high- and low-porosity structures are 325.68 MPa and 302.46 Pa, respectively, showing a decrease of 7.1% in the high-porosity structure. For the body-centered cubic porous structure, the residual stress after construction for the high- and low-porosity structures are 344.62 MPa and 371.99 Pa, respectively, showing an increase of 7.9% in the high-porosity structure. As indicated in Tables 7 and 8, the honeycomb porous structure has the smallest residual stress after construction, whether it is of low or high porosity.

## 5. Conclusions

The finite element simulation analysis of the renewable bone porous scaffold with the optimization of the additive manufacturing process provides a reference value and scientific guidance for the feasibility of porous humeral bone scaffolds, and it provides a basis for the research and design of clinical humeral bone scaffolds. We can derive the following conclusions:

1. A strut diameter that meets the manufacturing requirements of 3D printing technology was designed, followed by a pore size that can facilitate cellular transport of nutrients and metabolic waste. The porosity was also conducive to the growth of bone tissue into it, and four different types of unitary structures and finite element models of cuboid, diagonal, honeycomb, and body-centered cuboid were established;

2. Combining the four different porous structures with different porosities, compression simulations were carried out, and the stresses were increased in the high-porosity more than in the low-porosity structures. The cubic structure was the best in low-porosity structures, and the face-centered cubic structure was the best in high-porosity structures; the cubic structure was the best from the point of view of strain combined with deformation volume. For the influence of mechanical properties, porosity is the main factor;

3. Forming simulation analysis of porous structures shows that when fewer layers of suspended structures are constructed, the shrinkage force during the cooling phase is weakened by the limiting effect of the underlying powder, resulting in increased shrinkage. The energy density that can be absorbed by the metal material during construction easily induces a strong Marangoni effect, which can easily form cracks at the weld seam while adding to the risk of stress corrosion after implantation, as the human blood is weakly alkaline, which can accelerate stress corrosion near the weld seam;

4. The greater the height, the lower the stress level of the base plate itself and the more uniform the stress level of the part before dismantling. The stress distribution before dismantling includes large tensile stress zones in the upper area of the part being built. The maximum stress (equal to the yield stress) is reached at the surface of the part. The removal of the part significantly reduces the residual stresses present in the part; the residual stresses are relaxed by uniform shrinkage and bending deformation. The stresses in the part after disassembly are much less than before disassembly.

**Author Contributions:** H.M.: data curation, formal analysis, writing—original draft preparation and funding acquisition. S.X.: data curation and methodology. A.T.: conceptualization, methodology, review and editing, and funding acquisition. X.J.: investigation, methodology, and software. X.H.: investigation and software. All authors have read and agreed to the published version of the manuscript.

**Funding:** This study was financially supported by the National Natural Science Foundation of China (Grant No. 52275447), Key industrial projects to replace old and new driving forces in Shandong Province, China (New Energy Industry 2021-03-3), the Major Project of the Science and Technology Enterprise Innovation Program of Shandong Province, China (Grant No. 2022TSGC2108 and 2022TSGC2402), the Shandong Graduate Education and Teaching Reform Research Project (Grant No. SDYJG21169), the National College Student Innovation and Entrepreneurship Program (Grant No. 202210430010 and 202210430008), and the High quality curriculum construction project of Shandong Jianzhu University graduate education (YZKC202210 and ALK202210).

**Institutional Review Board Statement:** Not applicable.

**Informed Consent Statement:** Not applicable.

**Data Availability Statement:** The data presented in this study are available on request from the corresponding author.

**Conflicts of Interest:** The authors declare no conflict of interest.

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
