# Peer review of "Finite Element Analysis of Renewable Porous Bones and Optimization of Additive Manufacturing Processes"

_coatings, doi:10.3390/coatings13050912_

Round 1
Reviewer 1 Report
The work touches on a Finite Element Analysis that is relevant for biomedical industries, and the results obtained are of a certain scientific and applied value and interest. The manuscript can be recommended for publication after taking into account a number of comments.
1. What is the 3d printer that will be used to fabricate the porous structure from SS 316L?
2. Nothing is said about the 3D printing parameters to make the structure, since the structure is in micron scale therefore it is important to mention the printing parameters based on the literature review.
3. 3D printing is layer by layer manufacturing, is there any layer effect on the porous structure?
4. Conclusion is more like a discussion section. Please revise.
5. Line 505 "after implantation, the risk of stress corrosion increases due to the weak alkalinity of human blood.." any evidence on this statement?
Author Response
Thank the reviewer for your suggestions on this article. It is because of your suggestions that this article is more perfect and reasonable. We urge you to guide and evaluate the revised article again. We strongly hope that this article will be published after the revised, and thank you again for your efforts.
1.What is the 3d printer that will be used to fabricate the porous structure from SS 316L?
Response:Thanks to reviewers, the 3D printer used to manufacture porous structures in SS 316L is the FS121M metal additive manufacturing machine developed by China Huashu Hi-Tech Co. The authors have modified the paper accordingly.
2.Nothing is said about the 3D printing parameters to make the structure, since the structure is in micron scale therefore it is important to mention the printing parameters based on the literature review.
Response:Thanks to reviewers, the parameters of 3D printing equipment are: laser power 150W, scanning speed 500mm/s, scanning distance 0.06mm, powder layer thickness 0.035mm. The authors have modified the paper accordingly.
3.3D printing is layer by layer manufacturing, is there any layer effect on the porous structure?
Response:The reviewer's questions are good. 3D printing is manufactured layer by layer and the microstructure of multi-layered porous scaffolds based on laser melting forming is a fish scale microstructure similar to that of a weld pool, which is more homogeneous and free of layer effects through heat treatment.
4.Conclusion is more like a discussion section. Please revise.
Response:The author has streamlined and revised the conclusion section, which is more concise and clearer.
5. Line 505 "after implantation, the risk of stress corrosion increases due to the weak alkalinity of human blood.." any evidence on this statement? .
Response:The generation and extension of corrosion fatigue cracks in 316L stainless steel, a commonly used metal material for artificial joints, was studied in an artificial simulated body fluid (Hank's solution) using the applied potential method. The results show that the material has a tendency to porous corrosion in this solution.
Reference: Generation and extension of corrosion fatigue cracks in implanted 316L stainless steel in a human simulated environment, CHINESE JOURNAL OF BIOMEDICAL ENGINEERING, vol. 16, no. 3.277-288.
Reviewer 2 Report
To find the ways to prepare the porous metals to fit the bone tissue they replace in terms of mechanical compatibility, good adhesion for bone cells and allowing for the necessary nutrients and moisture to move through interconnected pores, is a challenging task of modern interdisciplinary medicinal science. In this regard, the main question addressed is to simulate using UG NX and Ansys the stress and strain for different porous structures with different parameters.
This is relevant topic to the field of coatings. Until now, no comparative study with the same technique to address the issue for different finite element and geometric models, namely, cubic, face-centered cubic, honeycomb, and body-centered cubic unit structures, is reported. In this sense, the present work addressing this issue and finding out the most suitable structures, fills the gap. On the basis of comprehensive modelling there are several new findings, such as that the design of the humeral bone scaffold as a porous structure could reduce the stiffness of the prosthesis, as well as alleviate stress shielding around the prosthesis after surgery and enhance its stability. In a certain sense, this is a novel scientific guidance for research and design of clinical humeral bone scaffolds.
The work would gain if, apart from the mechanical properties, the properties of the porous structures under investigation were studied in terms of dependence of diffusion/permeability parameters of liquids with different viscosity upon the specific structure (bcc, fcp, hpc) and porosity. Also, quite useful would be comparison of the modelling results with the experimental result for 316L prepared, for instance, by SLM, at least for one of the porous structures studied. At the same time these are subjects of another big studies. Conclusions are fully consistent with the evidence and arguments presented and they address the main issue of the work. Referencing is appropriate, as it fully meets the requirements, both in terms of the completeness of the coverage of the topic, and in taking into account the current state-of-the art in this area of research.
In general, the work is of high quality, is accurately conducted, clearly presented, relies on the relevant reference database, is within the scope of the journal Coatings. In my view, it is suitable for publication in the journal in its present form.
Author Response
Thank the reviewer for your suggestions on this article. It is because of your suggestions that this article is more perfect and reasonable. We urge you to guide and evaluate the revised article again. We strongly hope that this article will be published after the revised, and thank you again for your efforts.
Reviewer 3 Report
This paper did not address the scope of the Journal: coatings. However, paper is interesting, results using 3D printing, and additional modeling and simulation are very valyable.
Please, consider that Journal is related to coatings, and add some impact an usability of the obtained method and results to coatings application.
Table 1 and 2. Please explaine why 65% for low and 85% for high porosity are selected. are these porosities selected for similarity with bone sructures or for other reason.
Give explanation for Marangoni effect.
5. Conclusions: add one or two sentences at the beggining.
Author Response
1.This paper did not address the scope of the Journal: coatings. However, paper is interesting, results using 3D printing, and additional modeling and simulation are very valyable.
Response:Thanks to reviewers.
2. Please, consider that Journal is related to coatings, and add some impact an usability of the obtained method and results to coatings application.
Response:The author contributes to the album Recent Progress in Metal Additive Manufacturing, where the research on scaffolding covers Corrosion, Wear and Erosion.
3.Table 1 and 2. Please explaine why 65% for low and 85% for high porosity are selected. are these porosities selected for similarity with bone sructures or for other reason.
Response:Thanks to reviewers, the US Food and Drug Administration (FDA) recommends a porosity range of 35%-90% for porous femoral prostheses. Bragdon et al. suggest a porosity of at least 40% for porous structures that are adequate for bone growth. Simoneau et al. suggest a porosity of between 30%-50% for porous structures. Four different morphologies of porous structures were established and by adjusting the strut diameter and length, each structure was controlled for both high and low porosity, creating a contrast, with high porosity at around 86% and low porosity at around 65%. These porosities were chosen to be similar to the bone structure.
4.Give explanation for Marangoni effect.
Response:Thanks to the reviewer, marangoni effect: The phenomenon of mass movement due to a gradient of tension between the interfaces of two liquids with different surface tensions is known as the Marangoni effect.The authors have modified the paper accordingly.
5. Conclusions: add one or two sentences at the beggining.
Response:The author has revised inthe conclusions.
Reviewer 4 Report
In this manuscript the authors presented an interesting work on the Finite Element analysis of porous structures for bone tissue. The paper is well structured and the analysis is reported with high level of detail. My only concern is related to the mechanical simulation. Since the porous structures should be fabricated with an additive manufacturing approach, the anisotropy introduced with this fabrication method should be considered during the structure loading. Different behaviour should be highlighted (or not) according to different loading direction.
Other minor comments:
a, b, c, d labels are missing in Figure 1
Check labels position in all figures (should be under the single panel)
Line 201-203: please use SI units for weight
Author Response
In this manuscript the authors presented an interesting work on the Finite Element analysis of porous structures for bone tissue. The paper is well structured and the analysis is reported with high level of detail. My only concern is related to the mechanical simulation. Since the porous structures should be fabricated with an additive manufacturing approach, the anisotropy introduced with this fabrication method should be considered during the structure loading. Different behaviour should be highlighted (or not) according to different loading direction.
1. a, b, c, d labels are missing in Figure 1.
Response:The missing a, b, c and d labels have been modified in Figure 1.
2. Check labels position in all figures (should be under the single panel).
Response:The author checked the position of the labels in all figures and made changes.
3. Line 201-203: please use SI units for weight.
Response:Thanks to reviewers, The author has made changes in the manuscript.
Round 2
Reviewer 1 Report
The paper can be accepted as it is.
Reviewer 3 Report
Rersponse
1.This paper did not address the scope of the Journal: coatings. However, paper is interesting, results using 3D printing, and additional modeling and simulation are very valyable.
Response:Thanks to reviewers.
2. Please, consider that Journal is related to coatings, and add some impact an usability of the obtained method and results to coatings application.
Response:The author contributes to the album Recent Progress in Metal Additive Manufacturing, where the research on scaffolding covers Corrosion, Wear and Erosion.
Response given by authors did not give answers incorporated in text.
Rersponse
1.This paper did not address the scope of the Journal: coatings. However, paper is interesting, results using 3D printing, and additional modeling and simulation are very valyable.
Response:Thanks to reviewers.
2. Please, consider that Journal is related to coatings, and add some impact an usability of the obtained method and results to coatings application.
Response:The author contributes to the album Recent Progress in Metal Additive Manufacturing, where the research on scaffolding covers Corrosion, Wear and Erosion.
Response given by authors did not give answers incorporated in text.
Based on missing answers it is my opinion that in present state manuscript could not be published.